# Herbaceous Vegetation Responses to Gap Size within Natural Disturbance-Based Silvicultural Systems in Northeastern Minnesota, USA

**Nicholas W. Bolton [1]** and **Anthony W. D'Amato [2,*]**

[1]   Department of Forest Resources, University of Minnesota, St., Paul, MN 55108, USA; nwbolton@mtu.edu
[2]   Rubenstein School of Environment and Natural Resources, University of Vermont, Burlington, VT 05405, USA
*   Correspondence: awdamato@uvm.edu; Tel.: +1-802-656-8030

**Abstract:** The use of silvicultural systems that emulate aspects of natural disturbance regimes, including natural disturbance severities and scales, has been advocated as a strategy for restoring and conserving forest biodiversity in forests managed for wood products. Nonetheless, key information gaps remain regarding the impacts of these approaches on a wide range of taxa, including understory plant species. We investigated the 6- or 7-year response of herbaceous vegetation to natural disturbance-based silvicultural harvest gaps in a northern hardwood forest in Northeastern Minnesota. These results indicate that harvest gaps are effective in conserving understory plant diversity by promoting conditions necessary for disturbance-dependent understory plant species. However, harvest gaps also contained non-native invasive plant species.

**Keywords:** understory plant communities; natural disturbance-based silviculture; forest management; species conservation; northern hardwood forests

## 1. Introduction

The forest understory layer contains the majority of species richness within forest ecosystems around the globe [1]. Thus, conservation of understory plant species is an important goal of sustainable forest management [2], particularly in light of growing concerns regarding the loss of native biodiversity from managed systems [3]. Given the role natural disturbances play in the maintenance of biodiversity through effects on resource and propagule availability and microhabitat conditions [4–6], the use of silvicultural systems patterned after the severity and frequency of natural disturbances may serve as a management approach for restoring and maintaining native biodiversity within managed forests [7,8]. Nonetheless, evaluations of the response of understory vegetation community to these management regimes are largely lacking for most forest ecosystems [9–11] hampering our ability to develop systems for maintaining natural patterns of species richness and abundance within the understory layer.

Forest disturbance dynamics within northern hardwood forests in Northeastern North America are dominated by infrequent, low intensity tree fall gaps caused by wind, disease, and insects [12–15]. In many cases, these gaps provide opportunities for understory vegetation development not afforded by dense, close-canopied forest conditions due to the variety of resource conditions occurring across the gap environment [16–18] and the diversity of microhabitat conditions created by disturbance processes [19–21]. The response of understory vegetation to natural and harvest gaps of varying sizes has been well-studied [22–26]; however, the findings from this work are inconsistent. For example, several studies have documented differences in species abundance and composition across varying gaps sizes, whereas other work has found no difference in understory plant communities between

gaps and intact forest [27–31]. Reasons for these differences may include the varying gap sizes studied, the intensity of forest disturbance, and the condition of the forests studied (i.e., old-growth versus managed second-growth forests).

Inconsistencies between studies regarding the response of understory vegetation to canopy gaps has also been documented within a given forest type. For example, understory vegetation density differed between gaps and intact forest within northern hardwood forests of Northeastern USA [24]; however, investigations within the same forest type in the upper Great Lakes region have generated mixed results dependent on forest condition. In particular, no compositional differences between small gaps and intact forest conditions were detected in uneven-aged northern hardwood forests in the Upper Peninsula of Michigan [28], whereas species diversity was greater in canopy gaps relative to intact forests in second-growth stands in Northern Wisconsin [26]. Nonetheless, even where differences were not detected, Shields and Webster (2007) concluded that gaps provide opportunities for the regeneration of species absent from intact forest, particularly *Sanguinaria canadensis* L. (bloodroot), *Impatiens capensis* Meerb. (jewelweed), and *Rubus idaeus* L. (red raspberry). As such, the use of harvest gaps patterned after natural canopy gaps may provide an opportunity to enhance and restore understory plant richness in managed northern hardwood forest systems.

In many managed forests, management practices have historically focused on creating relatively uniform, homogeneous conditions with little diversity in the canopy, sapling, shrub, seedling, and herbaceous vegetation layers. Within the context of changing global conditions, these homogeneous systems are viewed as being highly vulnerable to future environmental changes [32]. Correspondingly, enhancing stand resilience in managed forest systems has become an emerging management objective [33]. In particular, management regimes that result in an increase in species and functional richness may improve a given forest's ability to adapt to environmental change [32,34]. In many cases, recommended management regimes for increasing adaptive capacity of managed systems to these changes have built on ecological silvicultural principles associated with creating a diversity of structural and compositional conditions through emulation of historic disturbance patterns [35]. This includes creating a range of canopy openings within the system to introduce spatial heterogeneity that may allow for the development of novel understory vegetation patterns on the landscape [2,31,36].

The present study examined the effects of harvest techniques that emulate natural gap openings on understory vegetation 6- or 7- years post-harvest within second-growth northern hardwood systems in the upper Great Lakes. The objective for this work was to develop an understanding of how understory vegetation responded to harvest gaps compared to the intact forest and the spatial distribution of understory vegetation within harvest gaps (i.e., gap edge and gap center). To achieve this objective, we examined the response of understory vegetation (herbaceous plant species and a singular shrub species (*Rubus idaeus*), to a range of gap sizes patterned after natural disturbances for the region [12]. We hypothesized that (i) large harvest gaps will enhance understory vegetation diversity and abundance relative to smaller canopy gaps and the intact forest and (ii) establishment patterns of understory vegetation will vary spatially across harvest gaps, particularly between gap edges and centers.

## 2. Materials and Methods

### 2.1. Study Sites

Study sites were located along the northern shore of Lake Superior in Northeastern Minnesota, USA (Table 1). Elevations within this area range from 381 to 472 m, and soils are loams derived from glacial tills [37]. Mean annual precipitation is 739 mm and annual temperatures range from −8.5 °C in January to 18.7 °C in July. Forests within the study area are dominated by northern hardwoods (Table 1) and scattered *Thuja occidentalis* L. (northern white cedar). Historically, the dominant overstory tree regeneration technique implemented in these systems were clearfelling. Understory vegetation within the study area includes *Clintonia borealis* (Aiton) Raf. (bluebead lily), *Streptopus lanceolatus* (Aiton) Reveal (twisted rosy-stalk), *Polygonatum pubescens* (Willd.) Pursh (Solomon's seal), *Claytonia caroliniana* Michx. (springbeauty), and *Thelypteris phegopteris* (L.) Sloss. (beech fern).

**Table 1.** Physiographic and compositional characteristics of the study sites in Northeastern Minnesota, USA.

| Site | Lat/Long | Harvest Year | Elevation (m) | Aspect | Slope | Soils | Canopy Composition [†] (% Basal Area) | Number of Gaps | Gap Size Range (ha) |
|---|---|---|---|---|---|---|---|---|---|
| Big Pine (BP) | (47.47, −91.15) | 2003 | 487 | 162° | 8% | Loam | *Acer saccharum* Marshall: 84% <br> *Betula alleghaniensis* Britt.: 3% <br> *Betula papyrifera* Marshall: 3% <br> *Thuja occidentalis* L.: 9% | 10 | (0.024–0.066) |
| Birch Cut (BC) | (47.45, −91.19) | 2002 | 455 | 119° | 6% | Loam | *Acer saccharum*: 89% <br> *Betula alleghaniensis*: 3% <br> *Acer rubrum* L.: 3% <br> *Picea glauca* (Moench) Voss: 3% <br> *Thuja occidentalis*: 3% | 10 | (0.021–0.071) |
| Power Line (PL) | (47.34, −91.20) | 2003 | 385 | 142° | 10% | Fine loam | *Acer saccharum*: 74% <br> *Fraxinus nigra* Marshall: 24% <br> *Populus grandidentata* Michaux: 2% | 10 | (0.011–0.069) |
| Schoolhouse (SH) | (47.46, −91.20) | 2002 | 478 | 327° | 2% | Loam | *Acer saccharum*: 63% <br> *Betula alleghaniensis*: 15% <br> *Picea glauca*: 15% <br> *Thuja occidentalis*: 7% | 16 | (0.008–0.050) |

[†] Canopy composition based upon surrounding intact forest data gathered by basal area swings with a 2.3 m$^2$/ha factor prism.

## 2.2. Study Design & Field Procedures

Harvest gaps were created within each site and replicated across four blocks in a completely randomized block design. Harvests took place during the winters of 2002 and 2003. Harvest gaps and intact forest conditions were measured in the summer of 2009 to assess the 6- and 7-year vegetation responses of these communities to harvesting treatments. To ensure adequate representation of gap environments, transects were laid across each gap oriented in subcardinal directions (northeast (NE), northwest (NW), southeast (SE), and southwest (SW)) and extended to the gap edge. Along each transect, 1 m$^2$ plots were systematically located and used for measuring tree regeneration and understory plant communities. Spacing between plots along each transect was adjusted according to gap size in order to provide an even distribution across each transect. To ensure consistent sampling intensities, each transect contained enough plots to represent 5% of each gap area (e.g., 8.1 m$^2$ plots for a 0.016 ha gap).

The location of each 1 m$^2$ plot within the gap was noted in the field to allow for analysis of the impact of spatial location on understory plant community composition. Plots falling within the outer third of a given gap were categorized as gap edge, and plots located in the inner third were categorized as gap center. The density (stems and clumps/m$^2$) and species of herbaceous understory plants, including herbs, ferns, fern-allies, graminoids, and *R. idaeus*, were recorded within each 1 m$^2$ plot. The inclusion of the woody shrub, *R. ideaus*, in our characterization of herbaceous understory plant communities was due to the recognized importance of this species in affecting vegetation development in canopy gaps in northern hardwood ecosystems [38,39]. An herbivory index was developed by dividing the number of *A. saccharum* Marshall stems with visible browse damage out of the first ten *A. saccharum* stems along a randomly chosen subcardinal transect within each gap or interior forest plot location. Subcardinal transects were also used for point-line sampling of coarse woody debris (CWD) volume estimates.

A series of control plots were placed in unharvested, intact portions at least 50 m from created gaps to serve as an approximation of pre-harvest vegetation conditions. For all control plots (*n* = 14), all measurements were performed in the same manner as study gaps. With the exception of Power Line (PL), each site contained three intact forest plots that were randomly located within unharvested portions of each stand. Due to a greater degree of variation in canopy composition at the PL site, an additional two plots were included within the intact forest portions of this site, resulting in five control plots at this site.

## 2.3. Statistical Analyses

Understory plant species densities were averaged for harvest gaps and the intact forest. Analysis of variance (ANOVA) was used to examine the impact of gap size on species richness and densities of understory vegetation [40]. ANOVAs were followed by Tukey-Kramer tests in cases in which significant gap size effects were detected. In addition, multivariate tests were conducted to assess understory vegetation compositional differences among gap size classes and spatial location using blocked multi-response permutation procedures (MRBPs) (PC-ORD version 5; McCune and Mefford 1999). Indicator species analysis was used to identify plant species most likely to be found within gap size classes and different gap locations (gap center, gap edge, or intact forest). Non-metric multidimensional scaling (NMS; [41]) was used to examine patterns in understory community composition within and among gap size classes. Due to differences in site conditions between blocks, NMS was run separately for each study site to allow for an evaluation of the effects of gap conditions on understory communities. Sørensen distances were used for MRBP and NMS to calculate a distance matrix for the harvest gaps and intact forest plots. To reduce noise in the data set, species with fewer than three occurrences were removed from the data matrices and the "slow-and-thorough" autopilot mode of NMS was used in PC-ORD to generate solutions [41]. Ordinations were run until a configuration of lowest stress was found, which for all four study blocks was a three-axis solution. The two axes that explained the most variability in the data are presented in the results section.

The relationship between understory community composition and environmental and forest structural characteristics (gap size, CWD volume, and herbivory index) were explored by using the bi-plot function overlaying a secondary matrix of these variables in PC-ORD [42]. Ordinations were rotated to place the environmental or forest structural variable with the highest correlation to understory community composition on the first axis. Relationships between species density and NMS axis scores were explored using Kendall's tau statistic in SAS [40].

## 3. Results

### 3.1. Density and Diversity of Understory Vegetation

Forty-two understory plant species or groups (e.g., *Carex* sp.) occurred across all study blocks. Thirty-three (79%) of those occurred across all harvest gap sizes and the intact forest. Understory vegetation was much denser in harvested gaps compared to intact forest (Figure 1), whereas richness was lower in small gaps compared to intact forest and was slightly higher in larger gaps (Figure 2). Certain understory species only occurred in gaps: *Actaea* sp. (baneberry), *Botrychium virginianum* (L.) Sw. (rattlesnake fern), *Mertensia paniculata* (Aiton) G. Don (northern bluebell), *Rubus idaeus* L. (red raspberry), *Sanguinaria canadensis* (bloodroot), and *Cirsium arvense* (L.) Scop. (Canada thistle). All other species occurred in both the intact forest and at least one harvest gap.

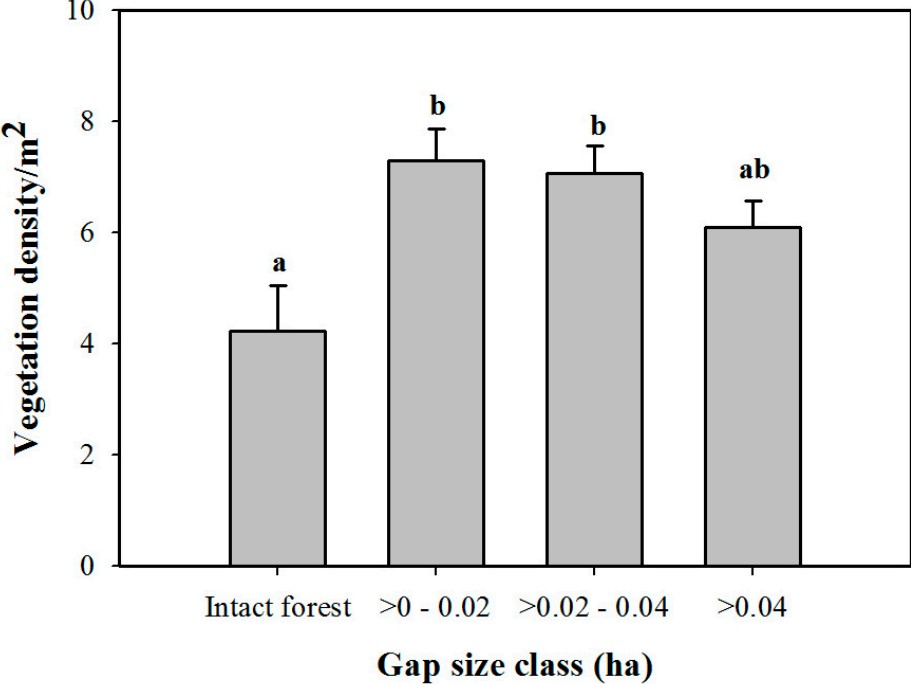

**Figure 1.** Understory vegetation density (stem count or clump per m$^2$) by harvest gap size (ha). Understory vegetation densities are averaged for all meter square sampling points throughout harvest gaps and intact forest plots. Error bars represent one standard error, and values with different letters are statistically different ($p \leq 0.05$; Tukey-Kramer test).

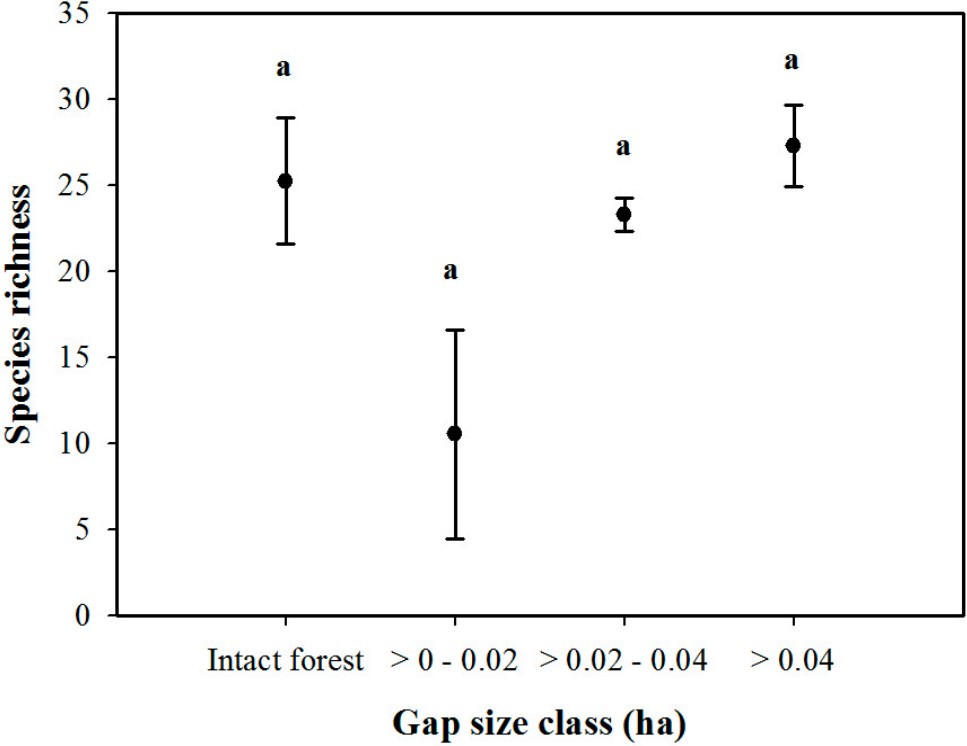

**Figure 2.** Understory vegetation species richness by harvest gap size (ha). All identified individual species encountered within all meter square sample plots were used to calculate richness. Error bars represent standard error. There were no significant differences between gap size classes ($p > 0.05$).

*3.2. Patterns in Understory Plant Composition across Gap Conditions*

Understory plant composition differed among gap sizes (MRBP; A-statistic = 0.017, $p$ = 0.0002). Species composition differed as gap size increased with the introduction of disturbance-adapted species, weedy plants, and non-native invasive species (Figure 3). In particular, abundances of *Botrychium virginianum*, *Cirsium arvense*, *Mertensia paniculata*, and *Rubus ideaus* increased as gap size increased. In contrast, there was no difference between understory plant communities within gap center and gap edge conditions. A significant indicator species ($p < 0.05$) for the intact forest was *Maianthemum canadense* Desf. (Canada mayflower). Whereas *Polygonatum pubescens* was indicative of gap edge conditions, and *Impatiens capensis* (jewelweed) and *Rubus idaeus* were indicative of gap center conditions, there was no statistical difference detected for understory plant composition between gap edge and gap center conditions. All ordinations had a final stress less than 15.5 and instability <0.0000001.

In addition to gap size being a significant environmental variable that explained the variability of the herbaceous communities within the study gaps, the herbivory index also explained the variability within experimental gaps. We did not specifically investigate the influence of browse on the herbaceous community response to these harvest gaps, but this result does indicate that larger harvest gaps were locations in which herbivores congregated.

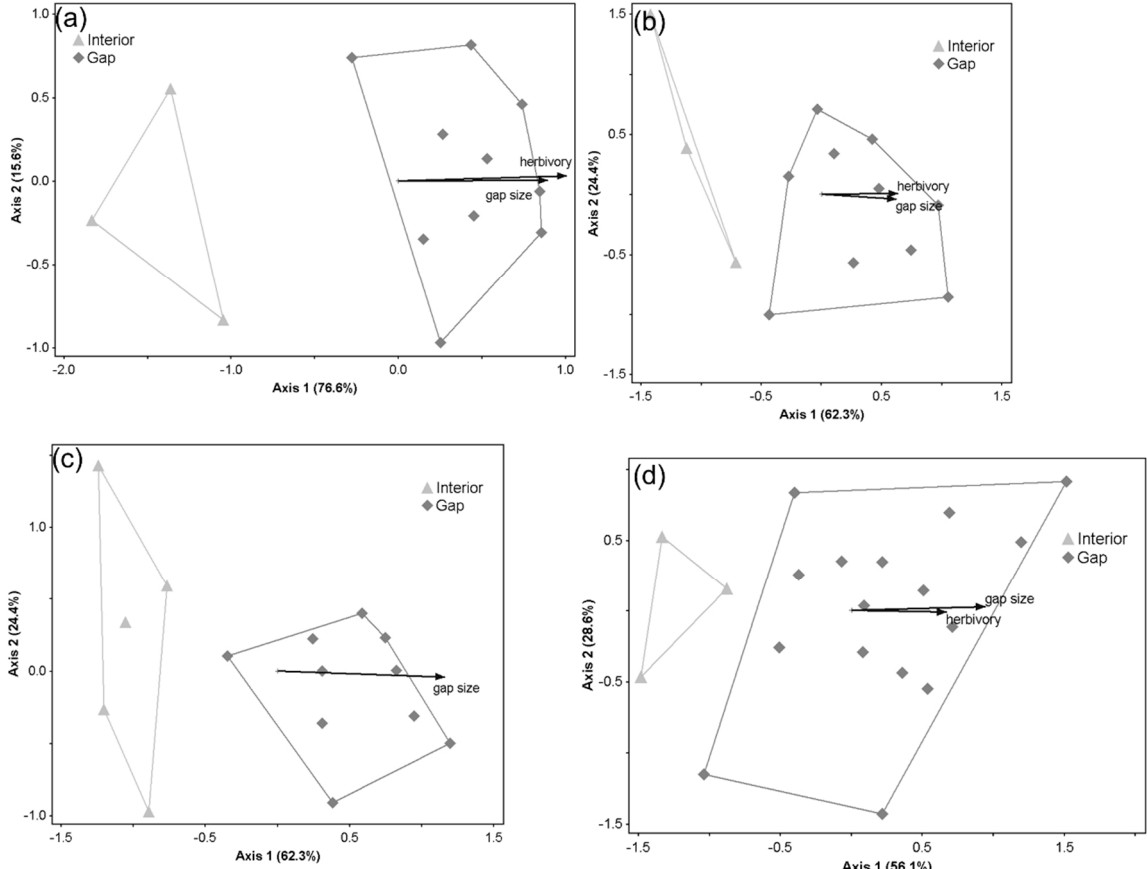

**Figure 3.** Nonmetric multidimensional scaling (NMS) ordinations of understory plant communities in gap and interior forest locations for the (**a**) Birch Cut, (**b**) Big Pine, (**c**) Power Line, and (**d**) Schoolhouse study areas. Vector length represents explanatory power of environmental variables (only variables with $r^2 > 0.2$ are depicted). Ordination diagrams were rigidly rotated to place the variable "gap size" parallel with the NMS axis 1. See Table 1 for study site information.

## 4. Discussion

*Understory Vegetation Responses*

Natural disturbances, including forest canopy gap formation, play an important role in structuring understory plant communities [23,43]. Correspondingly, the use of natural disturbance-based silvicultural (NDBS) regimes has been proposed as a means to maintain and restore native biodiversity to managed landscapes [8]. This study found that harvests patterned after natural canopy gaps increased understory vegetation richness within the northern hardwood systems examined—a result consistent with previous work examining a range of harvest gap sizes in the upper Great Lakes region [26,30]. In larger gaps (>0.02 ha), the occurrence of native species such as *Sanguinaria canadensis* increased. In addition, these gaps served to restore an element of spatial variability in the abundance of understory vegetation across these stands—a stand structural characteristic often lacking from second-growth northern hardwood stands [44].

Compositional changes between the intact forest and harvest gaps were largely due to the presence of disturbance-dependent species, wetland species, and non-native invasive species within harvest gaps. This finding is consistent with other studies following compositional changes post-harvest that have found that harvesting disturbance introduced weedy and non-native species resulting in increased species richness [27,45]. The primary disturbance-dependent species found within the stands examined was *Rubus idaeus*, which was found in greatest abundance in larger canopy gaps. This finding is consistent with other studies examining patterns of understory plant species in northern

hardwood forests [28,38,39]; however, work by Donoso and Nyland [38] suggests that the abundance of *R. idaeus* sharply declines in the decade following gap creation as canopy gaps close and advance regeneration overtop this species. Nonetheless, *R. idaeus* can hinder tree regeneration [38] within northern hardwood systems, particularly in areas with elevated herbivore densities, although we did not detect any evidence for this in the analysis of forest regeneration in these stands [46].

Although largely associated with wetter forest conditions, *Impatiens capensis* was found within canopy gaps within these stands. Similar patterns were documented by Shields and Webster [28] within canopy gaps in Northern Michigan, and they postulated that this phenomenon was due to increases in soil moisture stemming from the reduction of evapotranspiration following the removal of canopy trees. The presence of this species may also reflect its ability to establish and occasionally dominate areas following disturbance [47].

An increasing challenge within managed landscapes is the prevalence of non-native invasive species. In many cases, these species are associated with road edges, landings, and other highly disturbed locations [48]; however, many species are now being found within the interior of managed forests. Within this study, the non-native, invasive species *Cirsium arvense* was associated with large canopy gaps (>0.02 ha). The presence of this species in harvested stands has often been linked to the transport of seeds on logging equipment [49], as well as its ability to take advantage of soil compaction created by harvesting practices [50]. Given that the harvests in this study were done during the winter, it is likely that the main factors explaining the abundance of this species in larger gaps are its large dispersal range and aggressive establishment in disturbed areas [51,52].

The density of understory vegetation in harvest gaps was higher than the intact forest, a result similar to findings of other research in northern hardwoods [43,53,54]. Even small gaps had higher densities of understory vegetation per 1 m$^2$ compared to intact forest conditions, which is the opposite of the findings of Goldblum [24]. These differences are likely due to the difference in gap size classification (i.e., ranges of gap size classes) between the studies as well as time passed since disturbance. Gaps studied in Goldblum (1997) were 1–30 years post-disturbance, which are considerably longer time periods than the age of the gaps investigated in this study. Given these findings, it is possible that the changes in abundance of understory vegetation we documented within smaller gaps may be ephemeral and will decline as gaps close.

Most species encountered within the study gaps were present within the intact forest and increased in abundance as gap size increased. These species most likely took advantage of increased availability of resources within the gaps (e.g., light) [36,55,56]. Consistent with Shields and Webster [28] and Kern et al. [30], certain species that were not found within the intact forest were primarily established in large gaps, which included *Rubus idaeus*, *Impatiens capensis*, *Sanguinaria canadensis*, and *Cirsium arvense*.

Consistent with other studies within northern hardwood forests [9], the abundance of a few species did not change from the intact forest or across harvest gap sizes, including *Clintonia borealis* and *Aralia nudicaulis*. These species are rhizomatous [57] and primarily spread across the forest floor at very slow rates [57] as opposed to the species dominating the ground layer within the larger gaps we examined (e.g., *R. idaeus*), which are seed-bankers that often inhabit disturbed areas [58].

The lower diversity of understory plants found within small gaps (<0.02 ha) may be due to a combination of increased sapling growth rates from the harvest [46], damage to vegetation from logging operations [57], and harvest gap closures by border tree encroachment [59,60]. In particular, stimulated sapling growth and gap closure by bordering tree crowns likely minimized the amount of time resource levels were elevated at the forest floor layer following canopy gap formation. Although harvests led to an increase in the abundance of existing understory species (see above), there was limited opportunity for other species to regenerate, a finding consistent with research investigating gap partitioning in northern hardwood and western conifer forests (e.g., [30,61]). In this work, increases in species richness relating to resource availability were detected in larger gaps (e.g., >0.1 ha) within which resource gradients existed.

Research conducted in Northeastern Minnesota by Burton et al. [62] found that understory layers differed between second-growth forests managed by NDBS and old-growth forests. They concluded that second-growth forests similar to the ones examined in this study were higher in species diversity and contained higher levels of understory species abundance when compared to old-growth forests nearby. Given these findings, it is possible that the canopy gaps created in the current study are driving the structure and composition of the understory layers in these second-growth stands further from those found in old-growth forests. In particular, Burton et al. [57] speculated that the presence of a higher diversity of overstory tree species within old-growth stands, including *Betula alleghieniensis* Britt., *Picea glauca* (white spruce), and *Thuja occidentalis* (northern white cedar), resulted in a greater diversity of understory environmental conditions and higher levels of heterogeneity in understory plant communities. The gaps examined in this study, although containing increased levels of resource availability, largely released understory species that developed under a fairly mono-specific (i.e., *A. saccharum*) second-growth canopy. Future work aimed at restoring the diversity and structure of these communities may require a greater emphasis on restoring overstory community composition, particularly the currently underrepresented *B. alleghaniensis* and *T. occidentalis*.

## 5. Conclusions

The natural disturbance-based silvicultural treatment applied to the study area likely increased the richness of understory vegetation within the stand by providing opportunities for native species, disturbance-dependent species, and invasive species to establish. In particular, gap sizes between 0.008 and 0.07 ha allowed for the regeneration or establishment of several native species that were not found in the intact forest. However, the increase in richness also came at the expense of introducing aggressively establishing non-native invasive species and disturbance-dependent species into portions of these stands. The minimal gains in functional diversity need to be weighed in the context of the potential displacement of important native species by more aggressive native and non-native species. Much of the management within northern hardwood forests typically employs single-tree selection methods, which create relatively uniform canopy conditions and understory environments resulting in homogenization of the understory with relatively low species richness and understory structure [44]. This work adds to the growing body of work that suggests that the use of management regimes in which larger canopy gaps are created enhances compositional diversity of understory vegetation in second-growth northern hardwood forests of the upper Great Lakes, but may also introduce non-native invasive species if precautions are not taken [26,34]. Nonetheless, by creating a patchwork of high-density areas of understory herbaceous vegetation within gaps, these treatments restored some of the spatial diversity in understory abundance patterns found in old-growth northern hardwood systems [44].

**Author Contributions:** Methodology, analysis, writing, figure creation, and editing was conducted by N.W.B. & A.W.D.

**Funding:** This research was funded by University of Minnesota Department of Forest Resources and The Nature Conservancy. Access to the study locations was provided by the Lake County Minnesota Forestry Department.

**Acknowledgments:** We thank Meredith Cornett, Chris Dunham, Mark White, and Lee Frelich for their assistance during the experimental design, analysis, and reporting phases of this work. We appreciate Matthew Tyler for his vast knowledge of Minnesota native plants and field assistance.

**Conflicts of Interest:** The authors declare no conflict of interest.

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
