# Peer review of "Herbaceous Vegetation Responses to Gap Size within Natural Disturbance-Based Silvicultural Systems in Northeastern Minnesota, USA"

_forests, doi:10.3390/f10020111_

Reviewer 1 Report

I found the paper of general interest and well-written in the whole.  Nevertheless, there are some areas in need of improvement.  First, the justification for the work fails to cite a number of relevant recent papers, which gives the impression that the present study is among the first to consider these questions.  Also, in addition to omissions, some of the papers cited are not cited where I might expect in the text (e.g., introduction and lines 113, 119).  I’d suggest making a final pass to ensure that references are used effectively.

As far as omissions, I would point to some of the papers that have come out of the USFS Divide Gap Study and recent reviews:

Christel C. Kern, Rebecca A. Montgomery, Peter B. Reich, Terry F. Strong; Canopy gap size influences niche partitioning of the ground-layer plant community in a northern temperate forest, Journal of Plant Ecology, Volume 6, Issue 1, 1 February 2013, Pages 101–112, https://doi.org/10.1093/jpe/rts016

Christel C. Kern, Rebecca A. Montgomery, Peter B. Reich, Terry F. Strong; Harvest-Created Canopy Gaps Increase Species and Functional Trait Diversity of the Forest Ground-Layer Community, Forest Science, Volume 60, Issue 2, 19 April 2014, Pages 335–344, https://doi.org/10.5849/forsci.13-015

Christel C. Kern, Julia I. Burton, Patricia Raymond, Anthony W. D'Amato, William S. Keeton, Alejandro A. Royo, Michael B. Walters, Christopher R. Webster, John L. Willis; Challenges facing gap-based silviculture and possible solutions for mesic northern forests in North America, Forestry: An International Journal of Forest Research, Volume 90, Issue 1, 1 January 2017, Pages 4–17, https://doi.org/10.1093/forestry/cpw024

Christopher R. Webster, Yvette L. Dickinson, Julia I. Burton, Lee E. Frelich, Michael A. Jenkins, Christel C. Kern, Patricia Raymond, Michael R. Saunders, Michael B. Walters, John L. Willis, Promoting and maintaining diversity in contemporary hardwood forests: Confronting contemporary drivers of change and the loss of ecological memory, Forest Ecology and Management, Volume 421, 2018, Pages 98-108, ISSN 0378-1127, https://doi.org/10.1016/j.foreco.2018.01.010.

The fact that the present paper isn’t filling quite as wide of a gap in the literature as advertised, however, should not take away from the fact that it presents a well-replicated study which will help broaden our understanding and the range of systems considered.   The consistency of the present study with these other works is particularly interesting as are the departures.  

Second, some clarifications are needed regarding the presentation of results.  I am unsure what the authors mean by vegetation density/m2 (e.g., Figure 2).  I did not see that they counted stems and the numbers are too low for herb stem counts.  If this is cover in m2 per m2 it would reduce to a proportion.  Please clarify and define in the text and figure caption.  Also, on line 89-91 text more suited to the discussion is presented in the results.  Do you have any statistics you can reference to support this claim? How much variation did it explain?

Third, the presentation of authorities along with scientific names is inconsistent with many species lacking authorities (e.g., lines 72-82, 174).

Fourth, in-text citations are not always consistent and I noted some discrepancies in the literature cited.  For example on Line 53 and 142, Shields and Webster and Goldblum are presented in text with their respective years in parentheses.  Later they are presented as one might expect with the reference number after the author names (27 and 142, respectively).   I also noticed that some papers do not contain their full author list, is this journal style or an error?

Finally, I did not note them by line but the text contains some minor grammatical errors that should be corrected during a final edit.

Author Response

Dear Reviewer,

Thank you very much for your thoughtful and detailed review. This manuscript is better for your efforts. Please find below our responses to your comments.

I found the paper of general interest and well-written in the whole.  Nevertheless, there are some areas in need of improvement.  First, the justification for the work fails to cite a number of relevant recent papers, which gives the impression that the present study is among the first to consider these questions.  Also, in addition to omissions, some of the papers cited are not cited where I might expect in the text (e.g., introduction and lines 113, 119).  I’d suggest making a final pass to ensure that references are used effectively.

As far as omissions, I would point to some of the papers that have come out of the USFS Divide Gap Study and recent reviews:

Christel C. Kern, Rebecca A. Montgomery, Peter B. Reich, Terry F. Strong; Canopy gap size influences niche partitioning of the ground-layer plant community in a northern temperate forest, Journal of Plant Ecology, Volume 6, Issue 1, 1 February 2013, Pages 101–112, https://doi.org/10.1093/jpe/rts016

Christel C. Kern, Rebecca A. Montgomery, Peter B. Reich, Terry F. Strong; Harvest-Created Canopy Gaps Increase Species and Functional Trait Diversity of the Forest Ground-Layer Community, Forest Science, Volume 60, Issue 2, 19 April 2014, Pages 335–344, https://doi.org/10.5849/forsci.13-015

Christel C. Kern, Julia I. Burton, Patricia Raymond, Anthony W. D'Amato, William S. Keeton, Alejandro A. Royo, Michael B. Walters, Christopher R. Webster, John L. Willis; Challenges facing gap-based silviculture and possible solutions for mesic northern forests in North America, Forestry: An International Journal of Forest Research, Volume 90, Issue 1, 1 January 2017, Pages 4–17, https://doi.org/10.1093/forestry/cpw024

Christopher R. Webster, Yvette L. Dickinson, Julia I. Burton, Lee E. Frelich, Michael A. Jenkins, Christel C. Kern, Patricia Raymond, Michael R. Saunders, Michael B. Walters, John L. Willis, Promoting and maintaining diversity in contemporary hardwood forests: Confronting contemporary drivers of change and the loss of ecological memory, Forest Ecology and Management, Volume 421, 2018, Pages 98-108, ISSN 0378-1127, https://doi.org/10.1016/j.foreco.2018.01.010.

The fact that the present paper isn’t filling quite as wide of a gap in the literature as advertised, however, should not take away from the fact that it presents a well-replicated study which will help broaden our understanding and the range of systems considered.   The consistency of the present study with these other works is particularly interesting as are the departures.  

This is a great suggestion and we apologize fornot included these citations in the previous version of the manuscript.  We have now integrated these papers in several places in the introduction, as well as in the Discussion and Conclusions sections and make sure to indicate the present study contributes to the body of work established by these papers. 

Second, some clarifications are needed regarding the presentation of results.  I am unsure what the authors mean by vegetation density/m2 (e.g., Figure 2).  I did not see that they counted stems and the numbers are too low for herb stem counts.  If this is cover in m2 per m2 it would reduce to a proportion.  Please clarify and define in the text and figure caption.  Also, on line 89-91 text more suited to the discussion is presented in the results.  Do you have any statistics you can reference to support this claim? How much variation did it explain?

In section 1, line 16 we mention that understory plants were measured by stem count or clump/m2. The densities throughout the sites when averaged were quite low as indicated in Figure 1. We added clarifying text to the figure caption which clarifies this message, thank you for your comment.Thank you for pointing out thatour language may have been too strong regarding the correlation of our herbivory index and gap size. We edited this text to better reflect that result.

Third, the presentation of authorities along with scientific names is inconsistent with many species lacking authorities (e.g., lines 72-82, 174).

Thank you for your comment, we standardized our reporting of species scientific names.

Fourth, in-text citations are not always consistent and I noted some discrepancies in the literature cited.  For example on Line 53 and 142, Shields and Webster and Goldblum are presented in text with their respective years in parentheses.  Later they are presented as one might expect with the reference number after the author names (27 and 142, respectively).   I also noticed that some papers do not contain their full author list, is this journal style or an error?

Thank you for your comment regarding the citation presentation within the manuscript.In text citations and the reference list were updated and reviewed for appropriateness and consistency.

Finally, I did not note them by line but the text contains some minor grammatical errors that should be corrected during a final edit.

Yes, we attended to grammatical issues throughout the text.Primarily there was a decent amount of comma and semi-colon mis-use!

Reviewer 2 Report

This is a nicely replicated little study with some interesting results. It reports on the composition of forest understory vegetation in a range of harvested gaps made in (primarily sugar maple) second-growth forest in northeastern Minnesota. Ten gaps, representing a range of sizes from 0.008 to 0.071 ha, were created in each of four different forest stands. Although I found the use of herbaceous plant density rather than cover an unusual metric, apparently plants were sparse (Fig.1) and discrete enough to justify this approach. The authors addressed gap size influences using a combination of ANOVA and multivariate ordination techniques. The paper is well written, and the interpretations are appropriately cautious and based on the results presented.

There are, however, some gaps (pardon the pun!) in the Methods and the Discussion. The following is a list of line by line concerns, queries and suggestions.

Abstract: this should probably mention that results refer to sampling conducted 6-7 years after harvesting.

P. 1, line 42 seems to conflict with line 16 in the abstract: are gap size effects well studied or not?

P. 2, l. 45: please use separate references to indicate which ones showed differences, which ones did not

p. 2, l. 72: what is the history of this second-growth forest? Did it originate from clearfilling, approximately how many years previously? What was the previous forest?

p. 2, l. 77: replace Rubus idaeous parens with a comma

p. 2, l. 84: northwestern shore instead of northern shore?

p. 2, l. 87: please elaborate for an international readership – what “northern hardwoods” or refer to Table 1

Table 1: please insert horizontal lines to distinguish sites: breaks in canopy composition among sites are difficult to distinguish

p. 4, l. 14: I see no reference is made to gap position in terms of the more shaded (southern) vs. more sunlit (northern) sides of gaps; certainly this must affect gap edge effects, and should at least be acknowledged, even if not accounted for statistically. OR, are your gap positions sufficiently well geo-referenced that you can test for gap side influence?

p. 4, l. 15: stem densities only? No cover estimates? Was there a minimum size for counting plants/seedlings, how did you deal with multiple stems of grasses, ferns?

p. 4, l. 32: you probably mean “plant species densities” rather “vegetation densities” here

p. 4, l. 43: what reference for Sorensen distances

p. 4, l. 50: CWD is not defined, nor is its measurement mentioned in the Field Practices section. Likewise, how was “available light” measured, and with what sensor and units?

p. 6, l. 84: insert comma after conditions for clarity

p. 6, l. 87-91: why is there no statistical analysis of % herbivory by gap size vs. intact forest?

p. 7, l. 100-103: seems more appropriate to the Introduction

p. 8, l. 118: I suggest “regenerating trees” instead of “advance regeneration”

p. 8, l. 119-121: what is the connection between Rubus and herbivory?

p. 8, l. 137: comma instead of semi-colon

p. 8, l. 142-143: “… which includes considerably longer time periods and a wider time range …”

p. 8, l. 153-155: not a sentence, please rephrase

p. 8, l. 156-165: I am glad to see that the authors have addressed, however speculatively, the remarkably low diversity (i.e., why lower than in intact forest) of plant species encountered in the small gaps

p. 8, l. 157: seems to need a comma after [42]

p. 8, l. 162: comma, not semi-colon

p. 9, l. 171-172: is this really further from old-growth status? But aren’t old-growth forests more gappy than second growth?

p. 9, l. 173: [53] is not Burton et al.; should probably be [58]

p. 9, l. 195: remove comma

References: several inconsistent uses of upper case and italics; please check carefully.

Some examples (and there are others!)

Line 243: upper case and italics for Acer, upper case New England

Line 250: upper case Missouri Ozarks

Line 281: upper case and italics for Rubus

Line 289: PC-ORD should be all upper case

Line 324: upper case and italics for Rubus idaeous

Author Response

Dear Reviewer,

Thank you very much for your thoughtful and detailed review. Our manuscript is much improved by your efforts. Please see our specific comments below.

This is a nicely replicated little study with some interesting results. It reports on the composition of forest understory vegetation in a range of harvested gaps made in (primarily sugar maple) second-growth forest in northeastern Minnesota. Ten gaps, representing a range of sizes from 0.008 to 0.071 ha, were created in each of four different forest stands. Although I found the use of herbaceous plant density rather than cover an unusual metric, apparently plants were sparse (Fig.1) and discrete enough to justify this approach. The authors addressed gap size influences using a combination of ANOVA and multivariate ordination techniques. The paper is well written, and the interpretations are appropriately cautious and based on the results presented.

There are, however, some gaps (pardon the pun!) in the Methods and the Discussion. The following is a list of line by line concerns, queries and suggestions.

Abstract: this should probably mention that results refer to sampling conducted 6-7 years after harvesting.

Thank you for your comment. This was added to the Abstract

P. 1, line 42 seems to conflict with line 16 in the abstract: are gap size effects well studied or not?

In the abstract we highlight the key information gaps from the well-studied subject matter further discussed in the Introduction.

P. 2, l. 45: please use separate references to indicate which ones showed differences, which ones did not

Thank you for your comment. We separated out the references by differences and similarities. We also added current literature to boost the context of this work within more recently published research.

p. 2, l. 72: what is the history of this second-growth forest? Did it originate from clearfilling, approximately how many years previously? What was the previous forest?

Thank you for your comment, we added the following in the methods section describing the historical harvesting practice in the area. “Historically, the dominant overstory tree regeneration technique implemented in these systems were clearfelling.”

p. 2, l. 77: replace Rubus idaeous parens with a comma

Corrected.

p. 2, l. 84: northwestern shore instead of northern shore?

It is common to refer to this area of Minnesota as the north shore.

p. 2, l. 87: please elaborate for an international readership – what “northern hardwoods” or refer to Table 1

Added a reference to Table 1 to draw the reader’s attention to that listing of tree species

Table 1: please insert horizontal lines to distinguish sites: breaks in canopy composition among sites are difficult to distinguish

Added the horizontal line to break up the species listings of each study site.

p. 4, l. 14: I see no reference is made to gap position in terms of the more shaded (southern) vs. more sunlit (northern) sides of gaps; certainly this must affect gap edge effects, and should at least be acknowledged, even if not accounted for statistically. OR, are your gap positions sufficiently well geo-referenced that you can test for gap side influence?

The size of the gaps within this study are quite small. As such we attribute the lack of difference from southern plots to northern plots to the little to no difference in light regimes between the southern and northern edges of the study gaps.

p. 4, l. 15: stem densities only? No cover estimates? Was there a minimum size for counting plants/seedlings, how did you deal with multiple stems of grasses, ferns?

Since the study took place ecological studies have moved towards cover estimates rather than stem densities. When clumps were encountered, (e.g., sedges, grasses, ferns) stems were counted.

p. 4, l. 32: you probably mean “plant species densities” rather “vegetation densities” here

Yes, thank you for your edit as it provides clarity to this sentence.

p. 4, l. 43: what reference for Sorensen distances

Sorenson’s index is a common use for ecological community analyses with NMDS. Wecited McCune and Grace for our use of the Sorensen’s index.

p. 4, l. 50: CWD is not defined, nor is its measurement mentioned in the Field Practices section. Likewise, how was “available light” measured, and with what sensor and units?

Thank you for your comment. We added descriptions of coarse woody debris and how it was sampled in the methods section. Additionally, we removed the comments about available light as we did not further discuss this measurement in the results or discussion sections.

p. 6, l. 84: insert comma after conditions for clarity

Yes, done.

p. 6, l. 87-91: why is there no statistical analysis of % herbivory by gap size vs. intact forest?

We never intended to assess these differences in a statistical environment, only as explanatory variables to our non-parametric analyses.

p. 7, l. 100-103: seems more appropriate to the Introduction

We elect to leave this statement here to set the stage for the Discussion section as canopy gaps are the subject of our study and further drive the compositional matrix within these forested systems.

p. 8, l. 118: I suggest “regenerating trees” instead of “advance regeneration”

Advance regeneration is a common forestry term for natural regeneration that is shade tolerant. These individuals establish and persist in the often-low light condition of the understory until a canopy opening is created.

p. 8, l. 119-121: what is the connection between Rubus and herbivory?

We highlight that the study gaps had higher levels of herbivory compared to the surrounding intact forest. Therefore, we suggest that canopy gaps serve as areas for herbivores to congregate. This in tandem with the dense pulse of Rubus within gaps further hinders regeneration opportunities on the forest floor.

p. 8, l. 137: comma instead of semi-colon

Yes, updated.

p. 8, l. 142-143: “… which includes considerably longer time periods and a wider time range …”

Compared to the current text I’m concerned that the suggested language is redundant, thank you for your comment.

p. 8, l. 153-155: not a sentence, please rephrase

Thank you for your comment, this revision greatly improved the sentence.

p. 8, l. 156-165: I am glad to see that the authors have addressed, however speculatively, the remarkably low diversity (i.e., why lower than in intact forest) of plant species encountered in the small gaps

p. 8, l. 157: seems to need a comma after [42]

Yes, updated. Thank you.

p. 8, l. 162: comma, not semi-colon

Updated, thank you.

p. 9, l. 171-172: is this really further from old-growth status? But aren’t old-growth forests more gappy than second growth?

Thank you for your comment, we were addressing the introduction of invasive species, lack of diversity, and low CWD volumes indicating low forest structure.

p. 9, l. 173: [53] is not Burton et al.; should probably be [58]

Yes, thank you for your comment.

p. 9, l. 195: remove comma

Yes, done. Thank you for your comment.

References: several inconsistent uses of upper case and italics; please check carefully.

Some examples (and there are others!)

Line 243: upper case and italics for Acer, upper case New England

Line 250: upper case Missouri Ozarks

Line 281: upper case and italics for Rubus

Line 289: PC-ORD should be all upper case

Line 324: upper case and italics for Rubus idaeous

Thank you for your specific comments regarding the inconsistencies of the references section within the manuscript. We updated the in-text citations as well as the references library to be consistentand appropriate for the journal.